# Rotational Bloch Boundary Conditions and the Finite-Element Implementation in Photonic Devices

Zhanwen Wang [1], Jingwei Wang [1], Lida Liu [1] and Yuntian Chen [1,2,3,*]

1    School of Optical and Electronic Information, Huazhong University of Science and Technology, Wuhan 430074, China; u201914251@hust.edu.cn (L.L.)
2    Wuhan National Laboratory of Optoelectronics, Huazhong University of Science and Technology, Wuhan 430074, China
3    Optics Valley Laboratory, Wuhan 430074, China
*    Correspondence: yuntian@hust.edu.cn

**Abstract:** This article described the implementation of rotational Bloch boundary conditions in photonic devices using the finite element method (FEM). For the electromagnetic analysis of periodic structures, FEM and Bloch boundary conditions are now widely used. The vast majority of recent research, however, focused on applying Bloch boundary conditions to periodic optical systems with translational symmetry. Our research focused on a flexible numerical method that may be applied to the mode analysis of any photonic device with discrete rotational symmetry. By including the Bloch rotational boundary conditions into FEM, we were able to limit the computational domain to the original one periodic unit, thus enhancing computational speed and decreasing memory consumption. When combined with the finite-element method, rotational Bloch boundary conditions will give a potent tool for the mode analysis of photonic devices with complicated structures and rotational symmetry. In the meantime, the degenerated modes we calculated were consistent with group theory. Overall, this study expands the numerical tools of studying rotational photonic devices, and has useful applications in the study and design of optical fibers, sensors, and other photonic devices.

**Keywords:** rotational Bloch boundary; finite element method; group theory

## 1. Introduction

Bloch theorem is a fundamental theorem in condensate matter physics and describes how the wave function of electrons behaviors in the periodic potential. In optics or electromagnetism, electromagnetic waves propagate in a similar fashion in periodic structures, as such the Bloch theorem can be used to simplify the electromagnetic analysis of periodic structures and improve the numerical computation efficiency. Indeed, Bloch boundary conditions were first used to solve the scattering problem of two-dimensional grating [1,2] and were, subsequently, used to analyze the electromagnetic scattering characteristics of three-dimensional cavity array [3]. Nowadays, FEM combined with the Bloch boundary conditions is widely used for the electromagnetic analysis of periodic structures [4–9]. In addition to the scattering problem, Bloch boundary conditions were also used for mode analysis, such as R. L. Ferrari's work on mode analysis of two-dimensional periodic structures [10], C. Mias and R. L. Ferrari's work on mode analysis of three-dimensional periodic structures [11], and A. A. Tavallaee's work on evanescent mode analysis [12]. These works mainly exploited the Bloch boundary conditions derived from discrete translational symmetry. G. Garcia-Contreras et al. proposed $C_n$ FEM by taking advantage of the rotational Bloch boundary conditions (RBBC) obtained from discrete rotational symmetry, and applied it to mode analysis and degeneracy analysis of two-dimensional rotationally symmetric waveguides [13].

Despite the large efforts in applying Bloch boundary conditions on periodic optical structures with translation symmetry, the use of Bloch boundary conditions for rotational symmetric structure did not receive sufficient attention yet. For instance, optical fiber has rotational symmetry [14], while detailed analysis of the mode properties in optical fibers using FEM concerning discrete rotational symmetry is absent. Adam Mock and Paul Trader derived RBBC using $C_{nv}$ symmetry of photonic crystal fibers, and combined it with FDTD to analyze the modes of photonic crystal fibers [15]. In theory, this method can be extended to FEM for mode analysis of photonic devices with complex structures and rotational symmetry.

In this paper, we studied the RBBC and practical implementation in FEM for the vectoral wave equation. Our numerical approach was flexible, and can be used in the mode analysis of any photonic device with discrete rotational symmetry. Notably, our approach reduced the computational domain to a periodic unit of the original, thus increasing computation speed and reducing memory usage. We solved the eigenmodes for the rotational photonic structure based on the proposed Bloch boundary conditions using FEM, and the degenerated modes we calculated were consistent with group theory.

The paper is organized as follows: In Section 2, the RBBC is derived from group theory and is implemented in finite element electromagnetic computation using the commercial simulation software COMSOL. In Section 3, we illustrate the validity of our approach via two examples: (1) eigen-mode analysis in two-dimensional photonic crystal fiber mode and (2) eigen-frequency analysis in three-dimensional photonic crystal resonator. Finally, the conclusion is reached in Section 4.

## 2. Theoretical and Numerical Models

### 2.1. Group-Character Labeled Wave Wunction and Domain Truncation in Rotationally Symmetric Structure

In a Cartesian coordinate system, the eigenmode field in an inhomogeneous medium photonic device fulfills the following vector wave equation:

$$\nabla \times \mu_r^{-1}(\nabla \times \boldsymbol{E}) - k_0^2 \epsilon_r \boldsymbol{E} = 0, \tag{1}$$

where $\boldsymbol{E} = \boldsymbol{E}(x, y, z)$ is the vector electric field, $\mu_r$ is the relative magnetic permeability, $\epsilon_r$ is the relative electric permittivity, and $x, y, z$ are the coordinates in the Cartesian coordinate system. If $\mu_r$ and $\epsilon_r$ have $C_n$ symmetry, then $\mu_r(r, \phi, z) = \mu_r(r, \phi + \Lambda, z)$, $\epsilon_r(r, \phi, z) = \epsilon_r(r, \phi + \Lambda, z)$, where $\Lambda = \frac{2\pi}{n}$, and $r, \phi, z$ are the coordinates in the cylindrical coordinate system. According to symmetry properties of wave function associated with the rotationally symmetric structure [16],

$$P_{c_n} E(r, \phi + \Lambda, z) = E\left(c_n^{-1}(r, \phi + \Lambda, z)\right) = E(r, \phi, z), \tag{2}$$

where $c_n$ is a rotational symmetry operation contained in the $C_n$ group, $P_{c_n}$ is the operator corresponding to the rotational symmetry operation $c_n$.

Based on group theory, $P_{c_n} E(r, \phi + \Lambda, z) = \chi(c_n) E(r, \phi + \Lambda, z)$, where $\chi(c_n)$ is the character corresponding to the rotational symmetry operation $c_n$. Therefore, the Equation (2) can be written as:

$$E(r, \phi, z) = \chi(c_n) E(r, \phi + \Lambda, z) \tag{3}$$

Equation (3) establishes the relationship between the electric field within the $\phi \subseteq [0, \Lambda]$ region and the entire region, thereby it is only necessary to computer the electric field within the $\phi \subseteq [0, \Lambda]$ range to infer the electric field throughout the entire region under the corresponding symmetry operation. As a result, the periodic unit can also be used to analyze the eigenmode field. It should be noted that in Equation (1), the electric field $\boldsymbol{E}$ is in the Cartesian coordinate system, while in Equations (2) and (3), the electric field $\boldsymbol{E}$ is in the cylindrical coordinate system.

### 2.2. Group-Character Revised Weak Form Formulation in Rotationally Symmetric Structure

In the eigenmode problem of photonic devices, the weak form of FEM can be written as $\int_{\Omega} \left[ (\nabla \times \boldsymbol{W}) \cdot (\mu_r^{-1} \nabla \times \boldsymbol{E}) - k_0^2 \epsilon_r \boldsymbol{E} \cdot \boldsymbol{W} \right] dV + \int_{\Gamma_b} (\boldsymbol{W} \cdot n \times \mu_r^{-1} \nabla \times \boldsymbol{E}) dS = 0$, where $\boldsymbol{W}$ is trial function, $\Omega$ is the entire computational domain, $\Gamma_b$ is the boundary of $\Omega$. When the photonic devices possess the $C_n$ symmetry as shown in Figure 1, the weak form can be expressed in the following form by utilizing Equation (3):

$$\int_{\Omega_{cell}} \left[ (\nabla \times \boldsymbol{W}) \cdot \left( \mu_r^{-1} \nabla \times \boldsymbol{E} \right) - k_0^2 \epsilon_r \boldsymbol{E} \cdot \boldsymbol{W} \right] dV + \int_{\Gamma_{b,cell}} \left( \boldsymbol{W} \cdot n \times \mu_r^{-1} \nabla \times \boldsymbol{E} \right) dS = 0, \quad (4)$$

$$[E(r, \phi, z)]_{\Gamma_d} = \chi(c_n) [E(r, \phi, z)]_{\Gamma_s}, \quad (5)$$

where $\Omega_{cell}$ is a periodic cell, $\Gamma_{b,cell}$ is the outer boundary of $\Omega_{cell}$, $\Gamma_s/\Gamma_d$ is the source/destination boundary, $[E(r, \phi, z)]_{\Gamma_s/\Gamma_d}$ represents the electric field on the boundary $\Gamma_s/\Gamma_d$. In the $C_n$ group, $\chi(c_n)$ is group character and takes the form $e^{jm\Lambda}$, where $0 \le m \le n-1$ and m is an integer. In order to obtain the entire eigenmode while applying RBBC for mode analysis, it is necessary to calculate m from 0 to $n-1$ for all situations. By utilizing RBBC in FEM analysis, the computational domain can be decreased to $1/n$ of its original size, which reduces the number of meshes and the number of degrees of freedom, as well as the memory required for the solution and the speed of the solution.

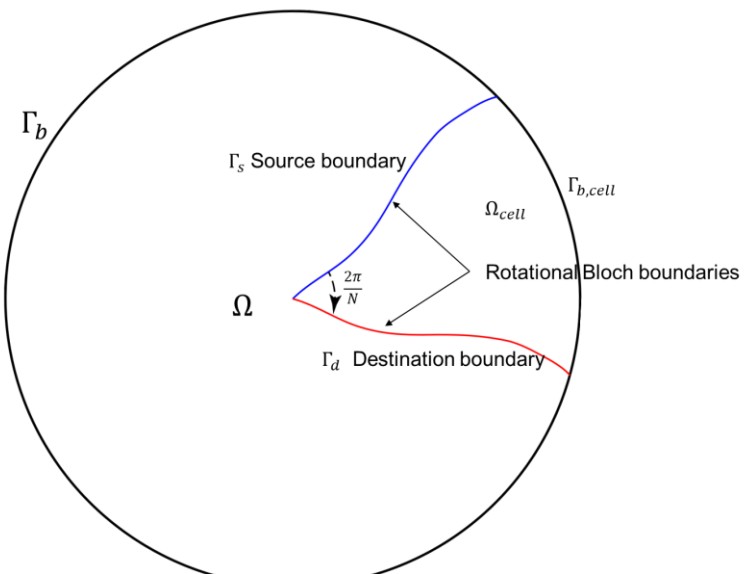

**Figure 1.** Physical Model.

### 2.3. COMSOL Implementation

We implemented the method in commercial numerical simulation software COMSOL in order to facilitate the use of the method. Equation (5) is in cylindrical coordinates, while the electric field in the frequency-domain electromagnetic wave component in COMSOL is in the Cartesian coordinate system. From Equation (5), the following Cartesian coordinate system-based periodic boundary conditions can be obtained:

$$
\begin{aligned}
\left[ E_x(x,y,z)\cos\phi + E_y(x,y,z)\sin\phi \right]_{\Gamma_d} &= \chi(c_n)\left[ E_x(x,y,z)\cos\phi + E_y(x,y,z)\sin\phi \right]_{\Gamma_s} \\
\left[ -E_x(x,y,z)\sin\phi + E_y(x,y,z)\cos\phi \right]_{\Gamma_d} &= \chi(c_n)\left[ -E_x(x,y,z)\sin\phi + E_y(x,y,z)\cos\phi \right]_{\Gamma_s} \quad (6) \\
\left[ E_z(x,y,z) \right]_{\Gamma_d} &= \chi(c_n)\left[ E_z(x,y,z) \right]_{\Gamma_s}
\end{aligned}
$$

In the practical implementation in COMSOL, Equation (6) is realized via the *Mapping()* function, which is used to bridge the relation between the electric field between $\Gamma_s$ edge and

$\Gamma_d$ edge as described in Equation (6). Specifically, the RBBC in COMSOL is summarized in Table 1. The third to fifth rows correspond to the three equations in Equation (6), such that all three electric field components meet the rotational Bloch boundary conditions. This technique was implemented by substituting the constraints and constraint forces in the periodic condition's component with the contents of Table 1.

**Table 1.** The implementation of constraints in COMSOL with RBBC-*FEM*.

| Rotational Bloch Boundary Conditions | Constraint | Constraint Force |
|---|---|---|
| $E_r(\phi + \Lambda) = \chi(c_n)E_r(\phi)$ | $Mapping(E_x\cos\phi + E_y\sin\phi)$ $= \chi(c_n)(E_x\cos\phi + E_y\sin\phi)$ | $test(Mapping(E_x\cos\phi + E_y\sin\phi))$ $= test(\chi(c_n)(E_x\cos\phi + E_y\sin\phi))$ |
| $E_\phi(\phi + \Lambda) = \chi(c_n)E_\phi(\phi)$ | $Mapping(-E_x\sin\phi + E_y\cos\phi)$ $= \chi(c_n)(-E_x\sin\phi + E_y\cos\phi)$ | $test(Mapping(-E_x\sin\phi + E_y\cos\phi))$ $= test(\chi(c_n)(-E_x\sin\phi + E_y\cos\phi))$ |
| $E_z(\phi + \Lambda) = \chi(c_n)E_z(\phi)$ | $Mapping(E_z) = \chi(c_n)E_z$ | $test(Mapping(E_z)) = test(\chi(c_n)E_z)$ |

*2.4. Finite Element Implementation*

In this work, we also implemented an alternative approach to realize the RBBC with a home-made code using MATLAB script. As for waveguide problems, our home-made code implemented FEM with our proposed RBBC; the procedures are discussed in details in the following. The transverse and vertical components of the electric field can be separated and written as follows

$$E(x,y,z) = E(x,y)e^{-\gamma z} = [E_t(x,y) + \hat{z}E_z(x,y)]e^{-\gamma z} \tag{7}$$

Here, $\gamma = \alpha + j\beta$ is the complex propagation constant where $\alpha$ and $\beta$ are respectively the real and imaginary parts of the complex propagation constant. $E_t$ represents the transverse component of the electric field, while $E_z$ represents the longitudinal component of the electric field. In order to facilitate the finite element method to solve the eigenvalue problem, the variable substitution is used to make $e_t = \gamma E_t, e_z = E_z$. The transverse component $e_t$ and the longitudinal component $e_z$ of the electric field are expanded on vector basis and scalar basis respectively as follows

$$\overline{e}_t = \sum_{j=1}^{m} \overline{e}_{t,j}\boldsymbol{\alpha}_j(x,y), \quad \overline{e}_z = \sum_{j=1}^{n} \overline{e}_{z,j}\alpha_j(x,y), \tag{8}$$

where $\overline{e}_t$ and $\overline{e}_z$ denote the numerical electric field, $\boldsymbol{\alpha}_j(x,y)$ denotes the vector basis functions implemented by the first type of Nédélec elements, $\overline{e}_{t,j}$ are the corresponding coefficients. While $\alpha_j(x,y)$ denotes the scalar basis functions using Lagrange elements and $\overline{e}_{z,j}$ are the corresponding coefficients.

Then, substitute Equations (7) and (8) into the weak form Equation (4). The final linear system of equations is of the form

$$\begin{bmatrix} 0 & 0 \\ 0 & \mu_r^{-1}[S_t] - k_0^2\epsilon_r[T_t] \end{bmatrix} \begin{bmatrix} \{\overline{e}_{z,j}\} \\ \{\overline{e}_{t,j}\} \end{bmatrix} = \gamma^2 \begin{bmatrix} \mu_r^{-1}[S_z] - k_0^2\epsilon_r[T_z] & \mu_r^{-1}[G]^T \\ \mu_r^{-1}[G] & \mu_r^{-1}[T_t] \end{bmatrix} \begin{bmatrix} \{\overline{e}_{z,j}\} \\ \{\overline{e}_{t,j}\} \end{bmatrix}, \tag{9}$$

where $S_{t,ij}^{(e)} = \iint_{\triangle_e}\left(\nabla_t \times \boldsymbol{\alpha}_i^{(e)}\right)\cdot\left(\nabla_t \times \boldsymbol{\alpha}_j^{(e)}\right)dS$, $T_{t,ij}^{(e)} = \iint_{\triangle_e}\boldsymbol{\alpha}_i^{(e)}\cdot\boldsymbol{\alpha}_j^{(e)}dS$, $S_{z,ij}^{(e)} = \iint_{\triangle_e}\nabla_t\alpha_i^{(e)}\cdot\nabla_t\alpha_j^{(e)}dS$, $T_{z,ij}^{(e)} = \iint_{\triangle_e}\alpha_i^{(e)}\cdot\alpha_j^{(e)}dS$ and $G_{ij}^{(e)} = \iint_{\triangle_e}\boldsymbol{\alpha}_i^{(e)}\cdot\nabla_t\alpha_j^{(e)}dS$. The corner mark $(e)$ denotes the grid element.

The eigenvalue problem of photonic devices in FEM can be expressed in a simplified form as follows

$$[A]\{x\} = \lambda[B]\{x\}, \tag{10}$$

where $\lambda = \gamma^2$ is the eigenvalue, $[A] = \begin{bmatrix} 0 & 0 \\ 0 & \mu_r^{-1}[S_t] - k_0^2 \epsilon_r[T_t] \end{bmatrix}$ and $[B] = \begin{bmatrix} \mu_r^{-1}[S_z] - k_0^2 \epsilon_r[T_z] & \mu_r^{-1}[G]^T \\ \mu_r^{-1}[G] & \mu_r^{-1}[T_t] \end{bmatrix}$ are the system matrix and $\{x\} = \begin{bmatrix} \{\bar{e}_{z,j}\} \\ \{\bar{e}_{t,j}\} \end{bmatrix}$ is the coefficient column vector of basis function. $\{\bar{e}_{z,j}\}$ and $\{\bar{e}_{t,j}\}$ contains the unknowns both on the boundary and inside.

As shown in Figure 2, the coefficients on the edge of the source boundary are denoted as $x_s$, those on the destination boundary edge are represented by $x_d$, and the rest of the unknown coefficients are denoted as $x_o$. Through Equation (5), we can obtain:

$$\begin{bmatrix} x_s \\ x_d \\ x_o \end{bmatrix} = \begin{bmatrix} I_S & 0 \\ \chi(c_n)I_D & 0 \\ 0 & I_O \end{bmatrix} \begin{bmatrix} x_s \\ x_o \end{bmatrix} \tag{11}$$

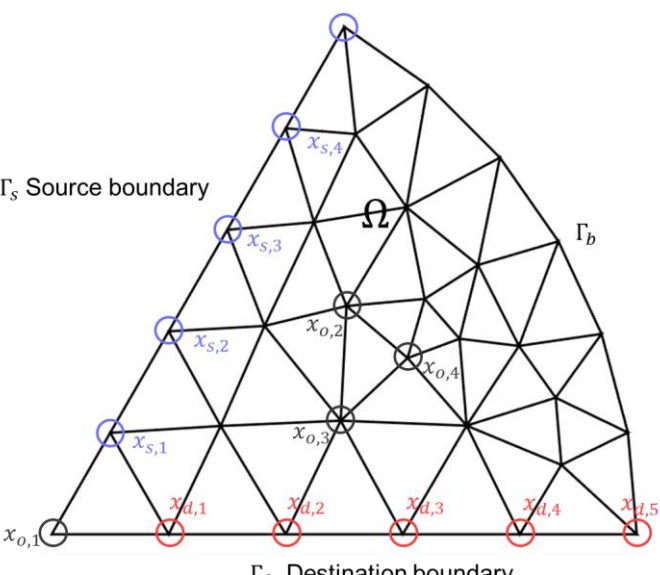

**Figure 2.** Rotational Bloch boundary conditions imposed.

The specific proof was given in Appendix A, where $I_S / I_D / I_O$ are identity matrices, and the subscript S/D/O represent the dimension of the coefficients on the source boundary/destination boundary/others. Equation (11) can be abbreviated as $\{x\} = [P]\{x'\}$, where $\{x\} = \begin{bmatrix} x_s \\ x_d \\ x_o \end{bmatrix}$, $\{x'\} = \begin{bmatrix} x_s \\ x_o \end{bmatrix}$, $[P] = \begin{bmatrix} I_S & 0 \\ \chi(c_n)I_D & 0 \\ 0 & I_O \end{bmatrix}$. As derived from $[P]^\dagger[A][P]\{x'\} = [A][x]$ [17], Equation (10) can be reformulated as:

$$[P]^\dagger[A][P]\{x'\} = \lambda[P]^\dagger[B][P]\{x'\}, \tag{12}$$

where $\dagger$ represents the conjugate transpose operator. Similarly, when solving Equation (12), we need to compute all cases where m is from 0 to $n-1$.

## 3. Results and Discussions

### 3.1. Two-Dimensional Photonic Crystal Fiber

In order to validate the accuracy of the proposed method, we verified it through two examples: one was the mode analysis in a two-dimensional photonic crystal fiber, the second example was the eigenfrequency analysis for a three-dimensional photonic crystal resonator. To make a fair compare in terms of accuracy and speed between con-

ventional boundary conditions (NBC) and RBBC in FEM, the mesh number for NBC was approximately $n$ times larger as that for RBBC.

In the first example, as shown in Figure 3, we considered the modal analysis for the photonic crystal fiber, the 2D cross section of which had $C_6$ symmetry. In the 2D plane, the spacing between the air holes was a, the fiber radius was 5.5a, the air hole radius was 0.3a. The operation wavelength λ was set to 1550 nm, and the width of the air ring around the fiber was 2λ. Silicon dioxide's refractive index was set at 1.45, relative magnetic permeability was 1. The boundary conditions were perfect electric conductors. The total number of meshes in the NBC-FEM was 62,604, whereas the total number of meshes in the RBBC-FEM was 10,415, with 48 eigenmodes calculated. The distribution of the electric field intensity in partial modes is shown in Figure 4, and it can be seen that the mode electric field intensity distribution and effective refractive index calculated by using NBC-FEM and RBBC-FEM implemented using COMSOL, as well as RBBC-FEM calculated in accordance with Section 2.4, had a close agreement. Due to the numerical singularity of the electric field at the tip, the mode with the effective refractive index of 1.3954 had a considerably larger error, necessitating an increase in mesh density at the tip when employing RBBC-FEM in practice. Other modes' effective refractive indices were consistent.

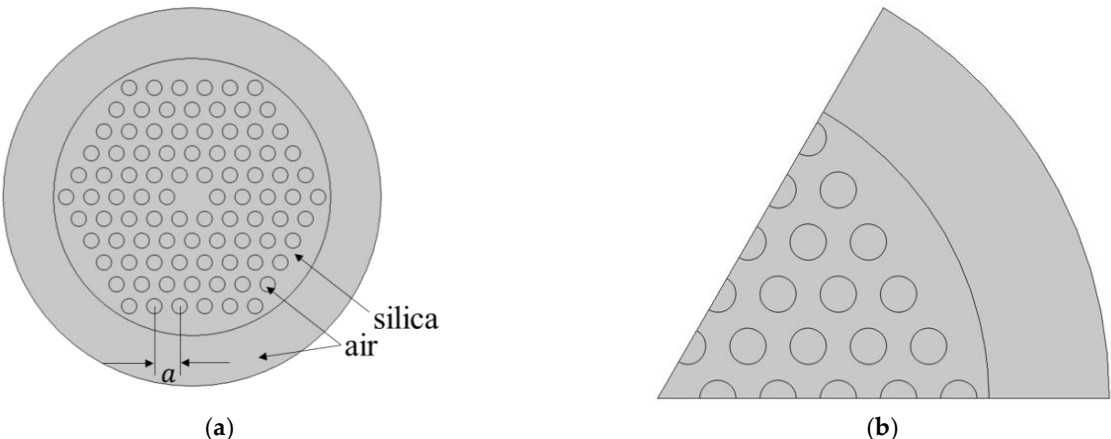

**Figure 3.** Two-dimensional photonic crystal fiber geometry. (**a**) NBC-FEM; (**b**) RBBC-FEM.

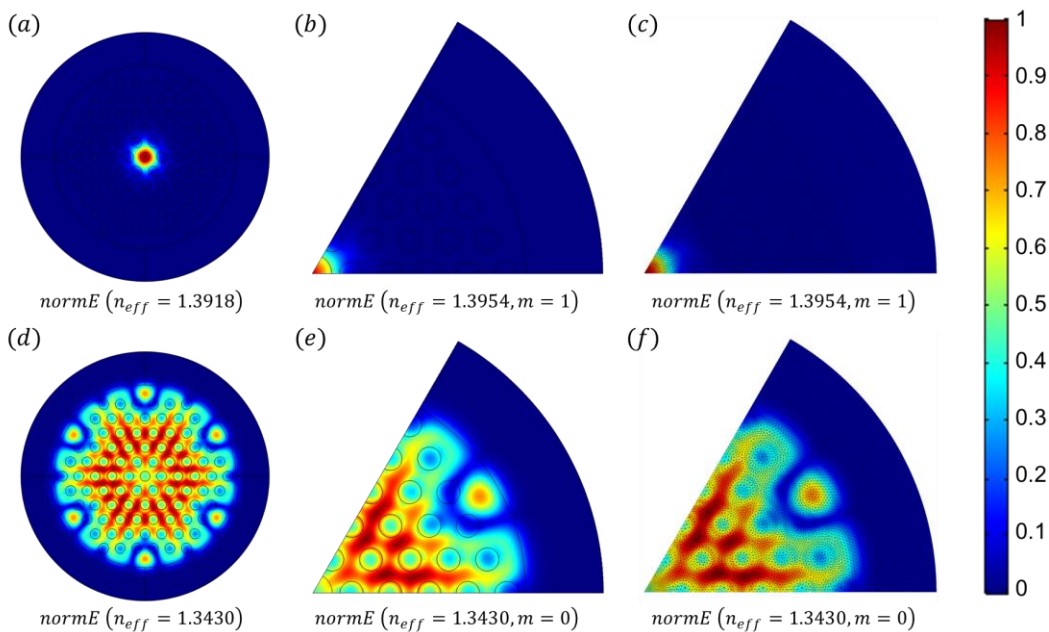

**Figure 4.** The mode analysis of two-dimensional photonic crystal fiber. (**a**,**d**) represent the NBC-FEM calculations from COMSOL, (**b**,**e**) represent the RBBC-FEM calculations from COMSOL, and (**c**,**f**) represent

the RBBC-FEM calculations from Section 2.4. (**a**–**c**) depict the electric field intensity distribution of the same base mode, with the NBC-FEM yielding an effective refractive index of 1.3918 and the RBBC-FEM yielding an effective refractive index of 1.3954, where m equals 1. (**d**–**f**) depict the electric field intensity distribution of the same higher-order mode, with the NBC-FEM and the RBBC-FEM both yielding an effective refractive index of 1.3439, where m equals 0.

Table 2 demonstrates the character table for the $C_6$ group, where $\chi(c_6) = \omega^m = e^{\frac{jm\pi}{3}}$ and $m \in [0, 1, 2, 3, 4, 5]$. The dual degenerate modes can be inferred from the two-dimensional irreducible representations $E'$ and $E''$ in the character table. These two two-dimensional irreducible representations were composed of two non-equivalent one-dimensional representations that were conjugate to each other. In the presence of time-inversion symmetry, the two non-equivalent one-dimensional representations that were conjugate to each other must be degenerated. Therefore, under the conditions that $m_1 + m_2 = 6$ and $\chi(c_n)$ is taken as $\omega^{m_1}$ and $\omega^{m_2}$ respectively, the computed modes will be degenerated.

**Table 2.** Character table for $C_6$ group.

| $C_6(6)\left(\omega=e^{j\pi/3}\right)$ | | | $E$ | $c_6$ | $c_3$ | $c_2$ | $c_3^2$ | $c_6^5$ |
|---|---|---|---|---|---|---|---|---|
| $x^2 + y^2, z^2$ | $R_z, z$ | $A$ | 1 | 1 | 1 | 1 | 1 | 1 |
| | | $B$ | 1 | $-1$ | 1 | $-1$ | 1 | $-1$ |
| $(xz, yz)$ | $(x,y)$ | $E'$ | 1 | $\omega$ | $\omega^2$ | $-1$ | $\omega^4$ | $\omega^5$ |
| | $(R_x, R_y)$ | | 1 | $\omega^5$ | $\omega^4$ | $-1$ | $\omega^2$ | $\omega$ |
| $(x^2 - y^2, xy)$ | | $E''$ | 1 | $\omega^2$ | $\omega^4$ | 1 | $\omega^2$ | $\omega^4$ |
| | | | 1 | $\omega^4$ | $\omega^2$ | 1 | $\omega^4$ | $\omega^2$ |

Table 3 shows the numerical results calculated using NBC-FEM and RBBC-FEM. The column "RBBC-FEM" represents the results obtained using RBBC-FEM, while the column "NBC-FEM" represents the results obtained using NBC-FEM. The second row shows the degree of freedom, and the number of degrees of freedom required by RBBC-FEM was only one-sixth of that required by NBC-FEM. The third row shows the memory usage, with RBBC-FEM consuming less memory. The fourth row shows the computation time, where RBBC-FEM's computation time for calculating all the six modes from m = 0 to m = 5 was only one-third of that of NBC-FEM. The reason that memory usage and computational time did not scale by a factor of six is that solving the generalized eigenvalue problem usually employs the Krylov subspace iteration method, which does not have a linear relationship with memory usage, computational time, and degrees of freedom, hence the absence of such scaling. As the modes calculated with m = 1 and m = 5 were degenerated, and those calculated with m = 2 and m = 4 were also degenerated, in practice, m only needed to be set to 0, 1, 2, and 3, which can further reduce the computation time to two-ninths of that required by NBC-FEM. The fifth row displays the effective refractive indices of the modes, which lists 33 modes in total. It can be observed that the error between the effective refractive indices calculated by RBBC-FEM and the NBC-FEM was within 0.3%.

Moreover, by comparing the effective refractive indices, it can be seen that the degenerate modes calculated by m = 1 and m = 5, as well as m = 2 and m = 4, was consistent with the analysis in the character table. By time reversal symmetry, the two one-dimensional representations that were each other's complex conjugate must necessarily be degenerated. In addition to degenerated modes, other high-order degenerate modes in RBBC-FEM can also be accurately calculated. These higher order degenerate modes are leaky modes and, consequently, have no physical meaning. They are provided in the table to illustrate the fact that RBBC preserved the generic rotational symmetry that guaranteed the possible degeneracy.

**Table 3.** A comparison table between RBBC-FEM and NBC-FEM calculations in COMSOL.

| FEM Class | $RBBC-FEM\left(E(r,\phi+\Lambda,z)=e^{im\Lambda}E(r,\phi,z)\right)$ | | | | | | NBC−FEM |
| | m=0 | m=1 | m=2 | m=3 | m=4 | m=5 | |
|---|---|---|---|---|---|---|---|
| DOF | | | 21,062 | | | | 125,589 |
| Memory | | | 1.67 GB | | | | 2.07 GB |
| CPU time | | | 13 s | | | | 36 s |
| | 1.3430 | | | | | | 1.3430 |
| | | | | 1.3451 | | | 1.3452 |
| | | | 1.3459 | | 1.3459 | | 1.3459(2) |
| | 1.3466 | 1.3466 | | | | 1.3466 | 1.3466(3) |
| | 1.3584 | 1.3584 | | 1.3585 | | 1.3584 | 1.3584(4) |
| | | | 1.3585 | | 1.3585 | | 1.3585(2) |
| $n_{eff}$ | 1.3678 | | | | | | 1.3678 |
| | | 1.3679 | | | | 1.3679 | 1.3679(2) |
| | | | 1.3680 | | 1.3680 | | 1.3680(2) |
| | | | | 1.3681 | | | 1.3681 |
| | 1.3784 | 1.3784 | 1.3784 | 1.3784 | 1.3784 | 1.3784 | 1.3784(6) |
| | 1.3880 | 1.3880 | 1.3880 | 1.3880 | 1.3880 | 1.3880 | 1.3880(6) |
| | | 1.3954 | | | | 1.3954 | 1.3918(2) |

If the time reversal symmetry is broken by changing Silicon dioxide to magneto-optical material with relative magnetic permeability $\begin{bmatrix} 1 & 0.51i & 0 \\ -0.51i & 1 & 0 \\ 0 & 0 & 1 \end{bmatrix}$, the modes with m = 1 and m = 5, as well as m = 2 and m = 4, will no longer be degenerated. Indeed, as shown in Table 4, as time-reversal symmetry was broken, the degenerated modes were split, though the RBBC still held. This was consistent with both simulation results and group theory analysis.

**Table 4.** A comparison table between RBBC-FEM and NBC-FEM calculations in COMSOL when the time-reversal symmetry is broken.

| FEM Class | $RBBC-FEM\left(E(r,\phi+\Lambda,z)=e^{im\Lambda}E(r,\phi,z)\right)$ | | | | | | NBC−FEM |
| | m=0 | m=1 | m=2 | m=3 | m=4 | m=5 | |
|---|---|---|---|---|---|---|---|
| DOF | | | 21,062 | | | | 125,537 |
| Memory | | | 1.52 GB | | | | 2.08 GB |
| CPU time | | | 16 s | | | | 47 s |
| | | | | 1.4364 | | | 1.4364 |
| | | | | | | 1.4374 | 1.4374 |
| | | 1.4395 | | | | | 1.4396 |
| | | | | | 1.4405 | | 1.4405 |
| | | | 1.4431 | | | | 1.4432 |
| $n_{eff}$ | 1.4437 | | | | | | 1.4437 |
| | 1.4456 | | | | | | 1.4457 |
| | | | | 1.4459 | | | 1.4459 |
| | | | | | | 1.4469 | 1.4470 |
| | | | 1.4476 | | | | 1.4477 |

Figure 5 illustrates the relationship between the effective refractive index and the wavelength of the fundamental mode calculated by RBBC-FEM and NBC-FEM. The solid blue line and the solid red dots represent NBC-FEM and RBBC-FEM with Silicon dioxide, respectively. The results they calculated were basically identical. Meanwhile, with the magneto-optical material, the results of RBBC-FEM, denoted by blue dashed line, and NBC-FEM, denoted by red hollow dot, were also exactly the same. It can be observed that the effective refractive index of the fundamental mode decreased as the wavelength increased,

which was in accordance with the optical fiber optics theory. Furthermore, RBBC-FEM's computation time shown in Tables 3 and 4 is the total computation time for calculating modes from 0 to 5. Only m = 1 needs to be calculated to get Figure 5. Therefore, RBBC-FEM far outperformed NBC-FEM at this time. Without breaking time-reversal symmetry, the calculation time of RBBC-FEM was 61 s, while NBC-FEM took 351 s. With breaking time-reversal symmetry, RBBC-FEM took 75 s and NBC-FEM took 391 s. RBBC-FEM's computation time was approximately one-sixth of that of NBC-FEM.

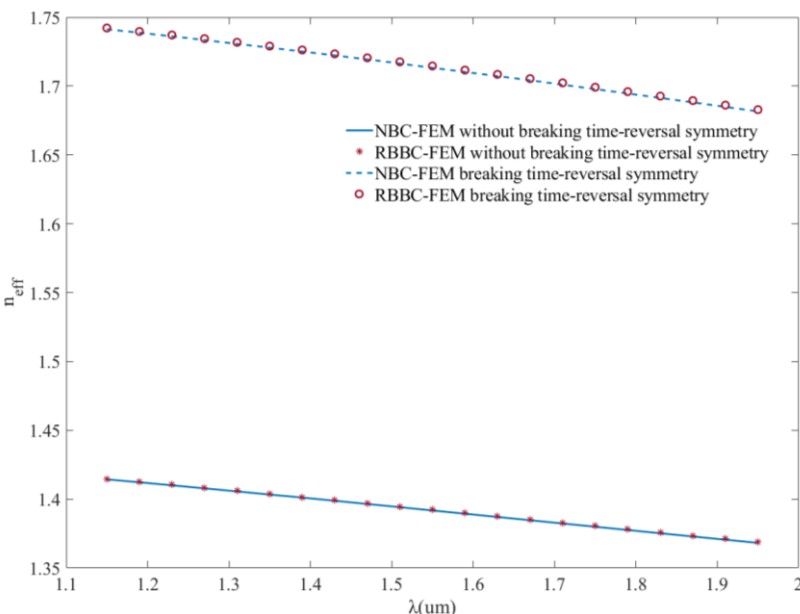

**Figure 5.** The relationship between the effective refractive index of the fundamental mode and the wavelength. Sweep wavelength from 1.15 um to 1.94 um, step length is 0.04 um.

### 3.2. Three-Dimensional Photonic Crystal Resonator

For the 3D photonic crystal resonator eigenfrequency analysis, the geometric structure was modeled using 1550 nm as the reference length, and the geometry is depicted in Figure 6, with the resonator radius of 5.5a, the air hole radius of 0.3a, and the spacing of a, a = 1550 nm, the ring width of the peripheral air ring being 2λ, a PML layer with a thickness of 2λ was added outside the air ring to simulate free space, and the resonator thickness h being 1 um. The index of refraction of silica was fixed to 1.45. The perfect electrical conductor was employed on both the upper and lower surfaces to make it a resonator. The NBC-FEM had a total of 1,404,107 meshes, while the RBBC-FEM had a total of 230,608 meshes, and 24 eigenmodes were determined.

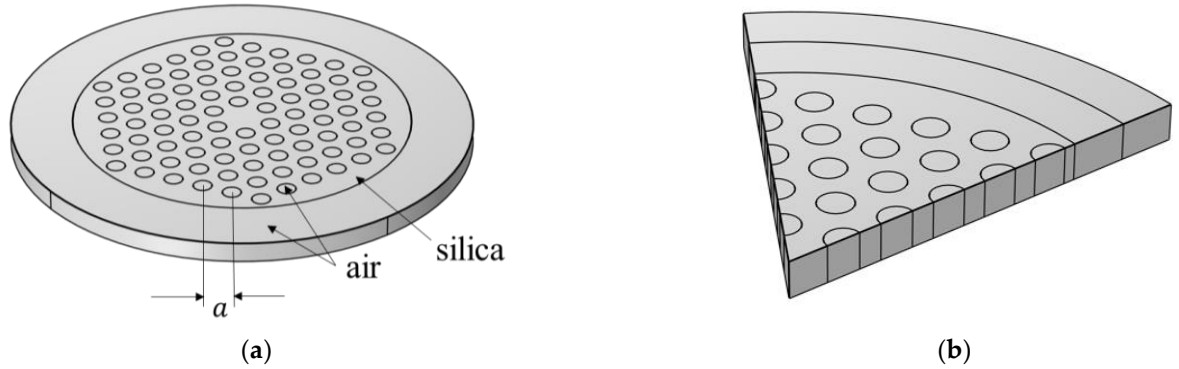

**Figure 6.** Three-dimensional photonic crystal resonator geometry. (**a**) NBC-FEM; (**b**) RBBC-FEM.

The electric field intensity distributions and eigenfrequency calculated from COMSOL using traditional NBC-FEM and RBBC-FEM are illustrated in Figure 7, which showed excellent agreement of the numerical calculations using two distinct boundary conditions in FEM, i.e., NBC-FEM and RBBC-FEM. Eigenfrequency analysis was conducted using COMSOL, and the outcomes of NBC-FEM and RBBC-FEM calculations are displayed in Table 5, which further confirmed the overall validity of our proposed RBBC-FEM against the NBC-FEM. The first column represents the RBBC-FEM results, whereas the second column represents the NBC-FEM data. The second row displays the degrees of freedom, and the needed degrees of freedom for RBBC-FEM are just one-sixth of those required for NBC-FEM. The RBBC-FEM consumed less memory than the NBC-FEM, as shown in the third column. The fourth column represents the computation time, while RBBC-FEM's computation time was only 4.4% of that of NBC-FEM. The RBBC-FEM and NBC-FEM consumed comparable memory resources in 3D problems due to two primary factors. Firstly, the memory occupation of RBBC-FEM was determined by the Krylov subspace iteration method, which did not have a linear relationship with degrees of freedom. Furthermore, because the computer memory we used was 64 GB, the memory occupation of NBC-FEM reached its maximum threshold apart from the impact of its solving method. The solution time depended on the Krylov subspace iteration method, and the degree of freedom of NBC-FEM was significantly larger than that of RBBC-FEM. When using the same level of subspace, NBC-FEM required more computational time for solving unless more memory was utilized. The complexity of the finite element solution for the three-dimensional problems increased dramatically as compared to the two-dimensional case, due to increased meshes and number of degrees of freedom. Notably, the solution time and the number of degrees of freedom did not have a simply linear relationship. In situations with significant large numbers of degrees of freedom, such as the three-dimensional numerical example with millions of degrees of freedom, the workload of the numerical solver will expand exponentially. Only the eigenfrequencies of the 19 modes are shown in the fifth row, which displayed the effective refractive indices of the eigenmode modes. By comparing the eigenfrequencies with the electric field intensity distribution, we can observe the degenerate modes calculated for m = 1 and m = 5, and for m = 2 and m = 4, which were consistent with the analysis in the character table and the two-dimensional scenarios.

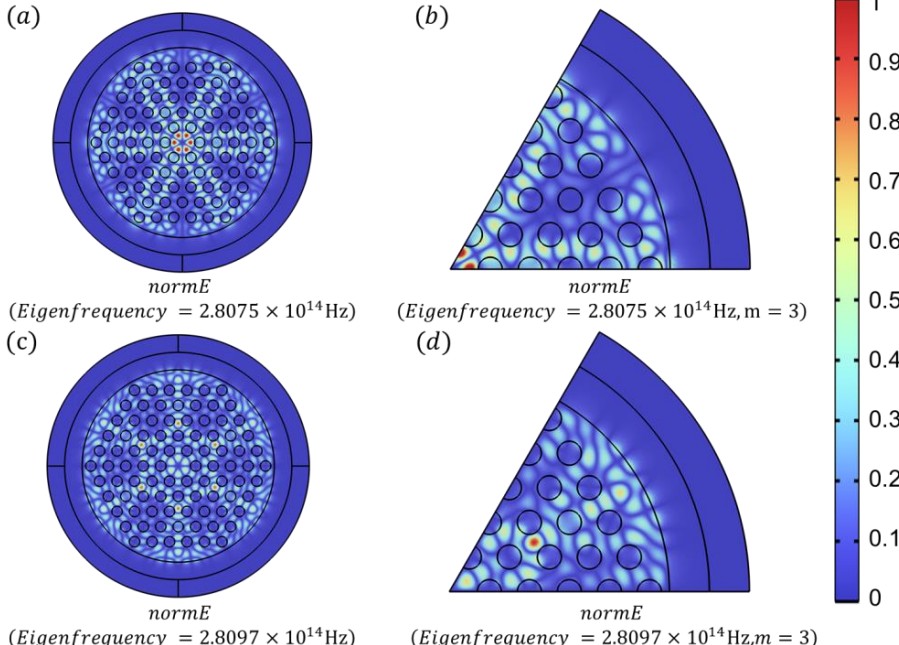

**Figure 7.** Computating eigenfrequency of a resonator in a three-dimensional photonic crystal structure. (**a**,**c**) Modal profile (field intensity) of Eigen-frequency at $2.8075 \times 10^{14}/2.8097 \times 10^{14}$ Hz calculated using

COMSOL NBC-FEM, (**b**,**d**) Modal profile (field intensity) of Eigen-frequency at $2.8075 \times 10^{14}$/ $2.8097 \times 10^{14}$ Hz calculated using COMSOL RBBC-FEM calculations. In (**a**–**d**), m is assumed to be 3.

**Table 5.** Comparison of RBBC-FEM and NBC-FEM 3D calculations in COMSOL.

| FEM Class | $RBBC-FEM\left(E(r,\phi+\Lambda,z)=e^{im\Lambda}E(r,\phi,z)\right)$ | | | | | | $NBC-FEM$ |
|---|---|---|---|---|---|---|---|
| | $m=0$ | $m=1$ | $m=2$ | $m=3$ | $m=4$ | $m=5$ | |
| *DOF* | | | 1,505,144 | | | | 9,022,532 |
| *Memory* | | | 44.83 GB | | | | 59.86 GB |
| *CPU time* | | | 18 min, 24 s | | | | 7 h |
| | | 2.7984 | | | | 2.7984 | 2.7984(2) |
| | | 2.8011 | | | | 2.8011 | 2.8011(2) |
| | | | | 2.8020 | | | 2.8020 |
| | | | | 2.8028 | | | 2.8028 |
| | 2.8031 | | | | | | 2.8031 |
| *Eigen-* | | | 2.8035 | | 2.8035 | | 2.8035(2) |
| *frequency* | 2.8042 | | | | | | 2.8042 |
| $\left(\times 10^{14} \text{ Hz}\right)$ | | | 2.8066 | | 2.8066 | | 2.8066(2) |
| | | | | 2.8075 | | | 2.8075 |
| | | 2.8078 | | | | 2.8078 | 2.8078(2) |
| | 2.8094 | | | | | | 2.8094 |
| | | | | 2.8097 | | | 2.8097 |
| | | | 2.8098 | | 2.8098 | | 2.8098(2) |

## 4. Conclusions

In summary, we presented a group theory-based FEM that may be efficiently used to perform mode analysis of photonic devices with $C_n$ symmetry. Group theory provides a solid framework that can be used to develop the RBBC for $C_n$ symmetric optical systems, which truncates the whole finite elements computational domain to a single periodic unit. We studied the practical implementation of these boundary conditions in the finite-element method and numerical simulation software COMSOL, explored its impact on degenerate modes, and illustrated it with examples of two-dimensional photonic crystal fiber and three-dimensional photonic crystal resonator. Moreover, we benchmarked our RBBC-FEM against the convectional FEM with whole computational domain, and found out that the calculated eigenmodes from the two independent FEM implementation showed perfect agreement and was consistent with group theory analysis. As for the FDTD approach, it was difficult to make a fair comparison in terms of calculation time and memory usage between FDE (Lumerical FDTD) and our method (RBBC-FEM) due to the poor performance of capturing the mode degeneracy in FDTD calculation (not show here); thus, fine grid was needed to clearly distinguish the true degenerate modes from numerical errors. The rationale of highly dense grid needed in FDTD was beyond the scope of our paper; thus, we confined the comparison our comparison within the existing FEM methods.

As an outlook, we envisaged that our work of applying RBBC in finite element computational photonics is useful for analysis and design of different photonic devices with rotational $C_n$ symmetry. By reducing the computational domain to periodic cells, the required degrees of freedom for the finite element solution were decreased significantly, thereby boosting solution efficiency. Importantly, the method relied solely on the $C_n$ symmetry of the device and can be applied to numerical computation and design of any photonic device with $C_n$ symmetry. Notably, the proposed approach can also be extended to scattering FEM problems, and in combination of various types spatial symmetries that are beyond $C_n$ symmetry. Thus, we believe that our work is useful and inspiring towards developing highly efficient FEM algorithms in computational photonics.

**Author Contributions:** Conceptualization, Y.C.; methodology, Z.W.; validation, Z.W., J.W. and L.L.; writing—original draft, Z.W., J.W., L.L. and Y.C. All authors have read and agreed to the published version of the manuscript.

**Funding:** National Key Research and Development Program of China (Grant No. 2021YFB2800303), National Natural Science Foundation of China (Grant No. 61405067) and the Innovation Project of Optics Valley Laboratory.

**Data Availability Statement:** The data presented in this study are available on request from the corresponding author. The data are not publicly available due to data sharing is not applicable to this article.

**Conflicts of Interest:** The authors declare no conflict of interest.

## Appendix A

In two dimensions, Equation (5) can be written as:

$$
\begin{aligned}
\left[E_r(r,\phi,z)\right]_{\Gamma_d} &= \chi(c_n)\left[E_r(r,\phi,z)\right]_{\Gamma_s} \\
\left[E_z(r,\phi,z)\right]_{\Gamma_d} &= \chi(c_n)\left[E_z(r,\phi,z)\right]_{\Gamma_s}
\end{aligned}
\tag{A1}
$$

The electric field in the Cartesian coordinate system and cylindrical coordinate system corresponds as follows: $E_z = E_z$, $E_r = E_x\cos\phi + E_y\sin\phi$. The unit tangential vector of the boundary $\Gamma_s/\Gamma_d$ is $\vec{e} = \{\cos\phi, \sin\phi\}$; thus, $E_r = \vec{e} \cdot E_t$. Equation (A1) can be written as:

$$
\begin{aligned}
\left[\vec{e}_d \cdot E_t(x,y)\right]_{\Gamma_d} &= \chi(c_n)\left[\vec{e}_s \cdot E_t(x,y)\right]_{\Gamma_s}, \\
\left[E_z(x,y)\right]_{\Gamma_d} &= \chi(c_n)\left[E_z(x,y)\right]_{\Gamma_s}
\end{aligned}
\tag{A2}
$$

where $\vec{e}_s/\vec{e}_d$ represents the unit tangential vectors of boundary $\Gamma_s/\Gamma_d$.

In FEM, the transverse and longitudinal components of the electric field are expanded using basis functions, $\bar{e}_t = \sum_{j=1}^{m} \bar{e}_{t,j}\boldsymbol{\alpha}_j(x,y)$ and $\bar{e}_z = \sum_{j=1}^{n} \bar{e}_{z,j}\alpha_j(x,y)$. $\boldsymbol{\alpha}_j(x,y)$ represents vector basis functions, $\bar{e}_{t,j}$ is the coefficient corresponding to the vector basis function, $\alpha_j(x,y)$ represents scalar basis functions, and $\bar{e}_{z,j}$ represents the corresponding coefficient of the scalar basis function. When $\boldsymbol{\alpha}_j(x,y)$ is chosen as the first type Nédélec elements, $\vec{e}_j \cdot \boldsymbol{\alpha}_j(x,y) = 1$, and $\vec{e}_j$ represents the unit tangential vector of the corresponding boundary, the physical meaning of $\bar{e}_{t,j}$ is the tangential component of the electric field on the boundary. It should be noted that, as shown in Figure A1, it is assumed that $\vec{e}_j$ is in the same direction as $\vec{e}_s/\vec{e}_d$ in this article. When $\alpha_j(x,y)$ is chosen as Lagrange element, $\bar{e}_{z,j}$ represents the value of $E_z$ at the node. Combining the finite element variables with the substitution of $e_t = \gamma E_t, e_z = E_z$, Equation (A2) becomes:

$$
\begin{aligned}
\bar{e}_{t,d} &= \chi(c_n)\bar{e}_{t,s} \\
\bar{e}_{z,d} &= \chi(c_n)\bar{e}_{z,s}
\end{aligned}
\tag{A3}
$$

Similarly, the electric field is expanded in three dimensions, $\bar{E} = \sum_{j=1}^{m} \bar{e}_j\boldsymbol{\alpha}_j(x,y,z)$, where $\boldsymbol{\alpha}_j(x,y,z)$ represents vector basis functions, $\bar{e}_j$ is the coefficient corresponding to the vector basis function. When $\boldsymbol{\alpha}_j(x,y,z)$ is chosen as the first type Nédélec elements, we have $\bar{e}_d = \chi(c_n)\bar{e}_s$.

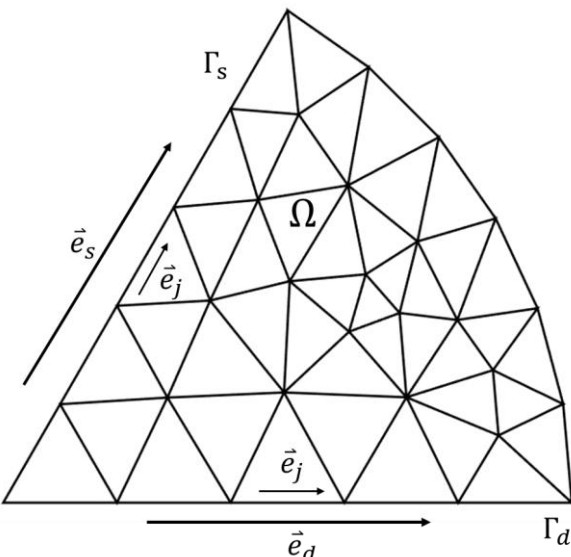

**Figure A1.** The direction of the boundary unit normal vector.

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
