# Peer review of "Rotational Bloch Boundary Conditions and the Finite-Element Implementation in Photonic Devices"

_photonics, doi:10.3390/photonics10060691_

Round 1
Reviewer 1 Report
In their manuscript, “Rotational Bloch boundary conditions and the finite-element implementation in photonic devices,” Wang and coauthors develop a Bloch boundary condition applicable to structures periodic in the azimuthal direction. Their derivation relies on rigorous group theory concepts, and the technique itself appears to be sound. However, I have significant concerns about the 2D and 3D photonic crystal fiber examples used to illustrate the method. Additionally, the benchmarking against similar FEM techniques appears to be inconsistent. I do not recommend publication unless the following issues can be addressed.
Line 75: The manuscript says Eq. 1 governs “the eigenmode field in a homogeneous medium photonic device”. Typically, photonic devices are not homogeneous media, so authors should be more specific about whether Eq. 1 applies to a homogenous medium only or to a photonic device which is spatially inhomogeneous.
Line 78: Manuscript says “epsilon_r is the relative electric conductivity”. This is incorrect; epsilon_r is the relative electric permittivity.
Line 164: “as well as RBBC-FEM calculated in accordance with Section 2.2”, does the manuscript mean “Section 2.4”? Same question for Fig. 4 caption (“Chapter 2.2”).
Manuscript lists “interpretation of the degenerated mode pairs based on group theory” as a novel contribution of the work in the abstract and introduction. But this concept is already well-known. The explanation of the origin of degeneracy is helpful in the discussion of the numerical results, but it is not itself new. I think the important point is that the proposed numerical technique obtains both the non-degenerate and degenerate modes as expected from group theory. I recommend to change the language accordingly.
Regarding computational resources for the 2d fiber, why do memory and CPU time not scale as a factor of six between RBBC and NBC? Also, for RBBC, does it take 13s for one m-value? If so, it takes 6x13=78 seconds for m=0-5 which is worse than the 36s for the NBC. Then for the 3d geometry, the memory scaling is similar to the 2d case, but the CPU time is significantly faster for the RBBC. Manuscript should discuss and clarify these issues more carefully.
Benchmarking should also be included for the “home-made code using MATLAB” (section 2.4).
About the two-dimensional fiber, the manuscript should show the fiber dispersion relation for the modes of interest. If this is a truly 2D simulation (which I suspect may be the case, correct me if I am wrong), then it is not really a fiber but rather a 2D simulation of a photonic crystal defect cavity (i.e. beta = 0). If that latter is correct, the language in the manuscript should be modified accordingly, and the relevant EM field vector components should be given.
Then moving to the 3D fiber, the role of the “peripheral air ring” should be explained. And what is meant by “cavity thickness h being 1 um”? Again, it seems this is not a fiber but rather a slab geometry. Manuscript should clarify.
Aside from a couple of awkward sentences, the English language is used well.
Reviewer 2 Report
This manuscript presents the implementation of rotational Bloch boundary conditions in photonic devices using the finite element method (FEM). By incorporating these boundary conditions, the computational domain is limited to one periodic unit, which improves computational speed and reduces memory consumption. The manuscript is generally well-written, but the following points should be addressed:
1. Could the authors compare their computational time and results obtained with other methods, such as the full-vectorial finite-difference method?
2. It would be interesting if the authors could demonstrate the application of their computational method in other photonic structures with high application value, such as micro-rings and micro-disks.
3. The principle of the FEM should be explained more clearly in the manuscript.
4. The full name of the FEM should be used when it first appears.
Round 2
Reviewer 1 Report
The authors have provided reasonable responses to most of my concerns. However, Section 3.2 continues to describe the 3d device as "Three-dimensional photonic crystal fiber resonator". This is factually inaccurate. The device is NOT a fiber. Rather it is a thin slab (the opposite of a fiber). The article should not be published unless Section 3.2 is revised accordingly.
It is also curious that for the 3d structure "the wavelength is set to λ=1550 nm", but in a resonator device, the geometry defines the resonance wavelength (it cannot be "set" externally). These statements in the manuscript make me question the soundness of all results in sections 3.1 and 3.2, but it may not be settled in this review process.
Reviewer 2 Report
The manuscript has been improved; however, I did not see clear advantages of their method over FDE in the paper. In the first question, the authors' response is somewhat vague. They did not explain the reason for choosing a mesh size of 1/30 of the wavelength in the FDE calculation, and it is unclear whether other mesh sizes, such as 1/20 of the wavelength, could be chosen instead. The authors did not provide any relevant data to support this choice. It is uncommon to select such a high mesh precision of 1/30 of the wavelength in FDE calculations.
Round 3
Reviewer 2 Report
The authors have addressed the reviewer's comments. So, this paper should be published in Photonics. However, there are still minor errors, such as the repetition of "it is" in line 321.
